# Personalized Low-Invasive Approach to Chronic Endometritis Evaluation in Premenopausal Women: Machine Learning-Based Modeling

**DOI:** 10.3390/diagnostics15222929

**Published:** 2025-11-19

**Authors:** Kseniia D. Ievleva, Alina V. Atalyan, Timur G. Baintuev, Iana G. Nadeliaeva, Ludmila M. Lazareva, Eldar M. Sharifulin, Margarita R. Akhmedzyanova, Leonid F. Sholokhov, Irina N. Danusevich, Larisa V. Suturina

**Affiliations:** Federal State Public Scientific Institution “Scientific Centre for Family Health and Human Reproduction Problems”, 664003 Irkutsk, Russia; alinaa@mail.ru (A.V.A.); tbaintuev@yandex.ru (T.G.B.); ianadoc@mail.ru (I.G.N.); lirken_@mail.ru (L.M.L.); sharifulja@mail.ru (E.M.S.); margarita.axmedzyanova@mail.ru (M.R.A.); lfshol@mail.ru (L.F.S.); irinaemails@gmail.com (I.N.D.); lsuturina@mail.ru (L.V.S.)

**Keywords:** chronic endometritis, machine learning, diagnostics, interleukins, leptin, adiponectin, androgens, C-reactive protein

## Abstract

**Background/Objectives**: Chronic endometritis (CE) is a well-known risk factor for recurrent implantation failure. However, the traditional approach to CE diagnosis has several drawbacks. On the other hand, there is a lot of evidence that some clinical, instrumental, and/or laboratory parameters of patients are associated with CE. The aim of this study is to build a CE prediction model using machine learning tools based on low-invasive pathological features. **Methods**: The data of 108 women (44 with and 64 without CE) from a multicenter perspective cross-sectional study was included in this study. Basic characteristics, reproductive history, laboratory and ultrasound indicators, and immunohistochemistry results were collected. Binary feature selection was performed using forward stepwise selection with logistic regression as the evaluation criterion. For each feature configuration, a gradient-boosting model was trained on decision trees with a binary logistic loss function. The models were evaluated and compared on test data using standard metrics. **Results**: We built five comparable predictive models for CE. The models yielded the following AUCs (95% CI): Model 1 (seven indicators)—0.704 (0.5170, 0.8907), Model 2 (seven indicators)—0.673 (0.4716, 0.8745), Model 3 (nine indicators)—0.677 (0.4916, 0.8622), Model 4 (five indicators)—0.758 (0.5913, 0.9241), and Model 5 (five indicators)—0.769 (0.5913, 0.9241). Models 2 and 5 have the better recall and precision values, but the differences were not significant. SHAP values indicated that serum adiponectin level (Model 2) and SHBG (Model 5) had the greatest association with CE risks. **Conclusions**: Models 2 and 5 show the most promising potential for clinical application, as they demonstrate superior recall and precision metrics and require assessment of only 5–7 risk markers (with only a few being non-routine) for their implementation.

## 1. Introduction

Chronic endometritis (CE) is strongly associated with recurrent implantation failure (RIF) [1]. Recent studies have consistently shown a significant prevalence of CE in infertile women, particularly among those diagnosed with RIF, recurrent spontaneous abortion, and unexplained infertility, with observed frequencies of 23.4%, 37.6%, and 19.46%, respectively [2,3]. On the other hand, Yilmaz et al. (2025) found that CE had been uncommon in patients with infertility and implantation failure. Consequently, routine diagnosis of CE in patients with these pathologies is not necessary, at least in the context of ART technologies [4].

Currently, the most specific method for evaluating CE is immunohistochemical examination of the endometrium for transmembrane heparan sulfate proteoglycan syndecan-1 (CD138). However, this method has several limitations. First, the techniques, conditions, and interpretations of immunohistochemistry for CD138+ in human endometrium have not yet been standardized. Second, the sampling methods used in endometrial biopsy are not fully effective and safe. Consequently, this procedure may fail to sample the required endometrial tissue, while potentially leading to the development of complications, such as endometrial thinning and intrauterine adhesions. Third, the diagnostic approach requires specific clinical indications. This often poses difficulties because of the asymptomatic or oligosymptomatic nature of CE [5]. In addition, patients have to endure a painful procedure.

These challenges highlight the need to develop less invasive and technically simpler methods for CE establishment. However, recent attempts to identify specific clinical or serum laboratory indicators of CE have been unsuccessful. Nevertheless, there is substantial evidence determining associations between certain clinical or laboratory indicators and CE.

Chen et al. (2016) [6] conducted a study involving 93 patients who had undergone laparoscopic and hysteroscopic examination for infertility. The research identified that the independent risk factors for CE had been episodes of prolonged menstrual bleeding, abortions, and fallopian tube obstruction. It is noteworthy that these patients exhibited no clinical signs of CE. Specifically, none of the patients reported pelvic pain, indicating the latent nature of the disease [6]. Hosseini et al. (2024) found an elevated risk of CE in women with endometrial polyps and uterine fibroids [7]. Kabodmehri et al. (2022) have shown a higher incidence of CE in patients with submucosal myoma than in patients with intramural and subserosal fibroids or in the control group [8]. In another study, researchers found a dependent relationship between CE and endometrial polyps (EPs): Women with EPs showed a higher prevalence of CE compared to women without EPs [9]. These findings suggest that ultrasound examination of pelvic organs and gynecological history data can also be used as part of a comprehensive approach to CE diagnosis. In our previous studies, we identified a significant association between elevated serum interleukin (IL) 1 levels, the IL-1/tumor necrosis factor α (TNFα) ratio, and adiponectin with CE in women with normal body mass [10,11]. Furthermore, low leptin levels were associated with CE in women who were overweight and obese [11].

Many of the described clinical and laboratory parameters have been shown to lack sufficient specificity for CE diagnosis. This may be related, among other factors, to the comorbidity associated with inflammation, including low-grade inflammation [12].

Therefore, we put forward a hypothesis: To enhance the diagnostic specificity and accuracy of CE evaluation based on low-invasive parameters, it is necessary to employ approaches that take into account the combinations of these parameters in a particular patient’s case. This novel diagnostic approach to CE has the potential to enhance early detection of this condition before complications and reproductive losses develop. Furthermore, the identification of CE in women with infertility will enable therapeutic adjustments aimed at restoring endometrial function. Previous studies have addressed CE prediction using deep learning models, primarily convolutional neural networks (CNNs), for analyzing hysteroscopic images to link findings with histopathologic CE diagnosis [13,14].

The aim of this study is to build a CE prediction model using machine learning (ML) tools based on low-invasive pathological features.

In this study, we present the results of developing extreme gradient-boosting machine learning models for predicting CE based on data from medical history, instrumental examinations, and laboratory tests of women from a non-selective population obtained during a previously conducted epidemiological study [15]. The results of the analysis suggest that a combination of certain clinical and laboratory parameters could be useful for CE establishment, thereby facilitating appropriate referral of patients for endometrial biopsy to confirm the diagnosis.

## 2. Materials and Methods

### 2.1. Patients

To conduct this analysis, we used the data from a multicenter perspective cross-sectional study of the prevalence of polycystic ovarian syndrome (PCOS) in an unselected multiethnic population of premenopausal women (ES-PEP study). This study included premenopausal women who were undergoing a mandatory annual employment-related health assessment, which ensured the non-selectivity of the sample. The database from this study contains information from a comprehensive examination, which comprised the following: questionnaire survey, general physical examination, gynecological examination, ultrasound examination, laboratory tests (biochemical, hormonal, and immunological), etc. [15]. We included 137 women in the analysis according to the following inclusion and exclusion criteria:

Inclusion criteria: age 18–45 years, and regular menstrual cycle.

Exclusion criteria: biochemical hyperandrogenism (HA) [16], hyperprolactinemia (prolactin ≥ 726 IU/L), hypothyroidism (thyroid-stimulating hormone [TSH] ≥ 4 mmol/mL), non-classic congenital adrenal hyperplasia (17-hydroxyprogesterone [17-OH] ≥ 6.9 nmol/L), premature ovarian failure (follicular stimulating hormone [FSH] ≥ 20 mME/L), amenorrhea, endometrial ablation, bilateral oophorectomy, sexually transmitted infections, and acute conditions.

Next, we excluded data of 29 women with body mass index (BMI) ≥ 30 due to the low quality of preliminary models, which could be associated with low-grade inflammation on the background of obesity. The final dataset included 108 women: 44 with CE and 64 without CE.

### 2.2. Clinical, Instrumental, and Laboratory Parameters

We included the following clinical, ultrasound, and laboratory parameters into the analysis: age, duration of menstrual cycle, visceral adipose tissue (VAT, %), menstrual cycle duration, number of days of heavy menstrual bleeding, Pictorial Blood Assessment Chart (PBAC), spontaneous abortion, extrauterine pregnancy, missed abortion, Cesarian section, endometrial thickness, uterine fibroids, endometrial polyp, serum level of C-reactive protein (CRP), anti-Mullerian hormone (AMH), total testosterone (T), sex hormone-binding globulin (SHBG), prolactin, TSH, 17-OH, FSH, luteinizing hormone (LH), estradiol (E2), leptin, adiponectin (ADIPOQ), IL-1, IL-4, IL-6, IL-8, IL-10, TNFα, and interferon (IFN) γ. Details of all measurements were previously described [10,15,17]. Also, we used calculated parameters: leptin/ADIPOQ and IL-1/TNFα ratio.

Evaluation of CE was conducted by expression of CD138+ in endometrial stroma and described earlier [17]. Briefly, on days 8–10 of the menstrual cycle, endometrial pipelle biopsies were performed. The obtained endometrial samples underwent immunohistochemical examination of CD138 expression using standard antibody kits (Dako, Glostrup, Denmark). The results were evaluated based on the presence or absence of a positive cytoplasmic reaction for CD138 in individual plasmatic cells within the endometrial stroma (plasma cells). According to the ES-PEP study protocol, an endometrial biopsy was performed on all participants who consented to the procedure, without selecting participants based on clinical symptoms of CE.

### 2.3. Machine Learning and Statistical Analysis Methods

#### 2.3.1. Data Pre-Processing

For this study, we use a dataset hosted in the IEEE DataPort repository: https://dx.doi.org/10.21227/7vd8-8f90 (accessed on 19 October 2025). This dataset contains raw, structured, and fully anonymized data in CSV format and includes all clinical, instrumental, and laboratory parameters used for ML and statistical analysis. The dataset is publicly available, ensuring the reproducibility of results and facilitating further research.

The data were checked for missing values and anomalies. Missing values were imputed using Multiple Imputation by Chained Equations (MICE) based on random forests. To address class imbalance in the training dataset, the Synthetic Minority Over-sampling Technique for Nominal and Continuous (SMOTENC) was applied. Numerical features were standardized using StandardScaler. The data was split into training (70%) and test (30%) sets while maintaining class proportionality.

Before imputation and standardization, the dataset was divided into two groups (with and without CE) and compared using the Mann–Whitney U test (for continuous variables) or the Chi-square test (for categorical variables) on the previously described parameters.

#### 2.3.2. Selection of Features and Model

Feature selection was performed using forward stepwise selection with logistic regression, with Receiver Operating Characteristic–Area Under Curve (ROC-AUC) as the evaluation criterion.

For each feature configuration, a gradient-boosting model was trained using decision trees with a binary logistic loss function, and the ROC-AUC metric was used as the evaluation criterion.

Given that we utilized the maximum possible exclusion criteria for sample formation (signs potentially associated with inflammation), we did not perform further confounding variable control.

#### 2.3.3. Model Evaluation and Interpretation

The models were evaluated on test data using the following performance metrics.

Accuracy (overall correctness of predictions), recall—the ability to correctly identify positive cases, precision—the proportion of true positive predictions among all positive predictions, specificity—the ability to correctly identify negative cases, F1-score—the harmonic mean of precision and recall, ROC-AUC—area under the ROC curve, and PR-AUC—area under the precision–recall curve. The following visualizations were generated: Confusion matrices to show classification results, ROC, and PR curves for precision vs. recall.

Model comparison was performed using the DeLong test [18] for ROC-AUC metrics and bootstrap resampling (1000 iterations) to calculate confidence intervals (95%) for PR-AUC and recall metrics. Differences were considered statistically significant at *p* < 0.05.

Feature importance and its impact on predictions were assessed using SHAP (Shapley additive explanations) values, including global interpretations through bar plots for feature importance visualization and Beeswarm plots for SHAP value distribution analysis. This approach provides a comprehensive understanding of how each feature contributes to the model’s predictions on both individual and aggregate levels.

To identify the most clinically applicable models, we relied on the following criteria:

Acceptable discriminatory power: AUC above 0.7 is generally considered acceptable, and an AUC of 0.6–0.7 may still provide useful discriminative capacity in many medical contexts [19,20].

Economic efficiency and minimal invasiveness: The feature sets in these models consist of variables that are low-cost and easy to collect, which is the cornerstone of our proposed diagnostic procedure.

#### 2.3.4. Software

Programming language: Python 3.12.2Runtime environment: Jupyter Notebook 6.5.4Libraries used: •Pandas 2.3.2—for structured data analysis and manipulation (DataFrame operations).•NumPy 2.2.0—for multi-dimensional array computations and mathematical operations.•Missingno 0.5.2—for visualizing missing data patterns in datasets.•Matplotlib 3.10.5—for creating basic static visualizations.•Seaborn 0.13.2—for advanced statistical plotting.•SciPy 1.16.1—for statistical analysis and mathematical functions.•Miceforest 6.0.3—for multiple imputation by chained equations (MICE) to handle missing values.•Imbalanced-learn 0.14.0—for handling class imbalance using the SMOTENC algorithm.•Scikit-learn 1.7.1—for data preprocessing, model building, and validation.•SHAP 0.48.0—for model interpretability and explanation of predictions.•Mlxtend 0.23.4—for machine learning tools, including feature selection.•XGBoost 3.0.4—for gradient-boosting machine learning models.•MLstatkit 0.1.9—for performing DeLong’s test to compare ROC-AUC of two models.

The source code developed for data preprocessing, selection of features, model training and model evaluation, and interpretation is publicly available in the associated GitHub 0.1.0 repository: https://doi.org/10.5281/zenodo.17347044 (accessed on 19 October 2025).

## 3. Results

### 3.1. Characteristics of Patients

We found that women with CE were older than women without CE, but this difference was not significant (*p* = 0.05). Women with CE had significantly lower levels of serum T, leptin/ADIPOQ ratio, TNFα, and IL-1/TNFα ratio, and had significantly higher levels of serum IL-1 (Table 1).

### 3.2. Prediction Models

Using logistic regression, we built a series of prediction models. We found that models built of data from gynecological history had a low quality due to the small number of cases (Appendix A). Consequently, we removed the gynecological history data from the dataset.

We built five distinct predictive models for CE whose metrics are presented in Table 2.

The SHAP summary plots provided a comprehensive visualization of feature effects on the predictive model performance, as illustrated in Figure 1. The features are ranked according to their average absolute SHAP values, from highest to lowest significance.

In the Model 1, low levels of TNFα, CRP, and T, as well as elevated ADIPOQ, IL-1, IFNγ, and the number of heavy menstrual bleeding days, were associated with a higher risk of CE. Due to the fact that we previously found out that IL-1/TNFα ratio had contributed more significantly to CE than IL-1, we replaced IL-1 and TNFα with the IL-1/TNFα ratio. Thus, in Model 2, elevated levels of ADIPOQ, 17-OH, E2, and VAT, as well as low T, IL-8, and CRP, were associated with a higher risk of CE. We found that the IL-1/TNFα ratio did not have enough impact on the prediction of CE. Next, we substituted individual measurements of leptin and ADIPOQ with their ratio based on compelling evidence that this ratio serves as a promising biomarker not only for metabolic disorders and low-grade inflammation but also for assessing endometrial receptivity [21]. In Model 3, elevated 17-OH, IL-1, E2, IL-6, PBAC, and heavy menstrual bleeding days, as well as low TNFα, T, and CRP, were associated with a higher risk of CE. To investigate the contribution of both relative characteristics, we built Model 4. In this Model, high-level SHBG and IFNγ, as well as low FSH, CRP, and leptin/ADIOQ ratio, were associated with a higher risk of CE. As our aim was to use for the prediction of CE not only laboratory but also basic and ultrasound low-invasive features, we conducted manual parameter-tuning methods to build Model 5. The last model included five parameters, from which elevated SHBG and low FSH, CRP, leptin/ADIPOQ ratio, and endometrial thickness were associated with a higher risk of CE. All CE prediction models demonstrated moderate quality according to the ROC-AUC value (Table 2).

### 3.3. Model Evaluation and Interpretation

Next, we compared all prediction models to find which one had the best quality and power of prediction. According to the DeLong test, all models had comparable ROC-AUC (with Bonferroni correction α = 0.0056) (Table 3, Figure 2a).

To evaluate the predictive performance of the models, we compared the recall and precision metrics across all models to identify the one demonstrating the highest quality and predictive power (Table 4, Figure 2a). Recall represents the proportion of truly positive cases (CE patients) correctly identified among all actual positive cases in the population. Precision, on the other hand, represents the proportion of correctly identified positive cases (true CE patients) among all cases classified as positive by the model. This precision analysis enables the identification of true cases while minimizing the misclassification of healthy subjects.

Bootstrap resampling analysis did not reveal statistically significant differences between the models (Table 4, Figure 2c,d). However, Models 2 and 4 were identified as the most promising for clinical application due to their superior recall and precision values (Figure 2a), as well as their superior ability to accurately predict true CE cases compared to the other models (Figure 2b).

## 4. Discussion

To the best of our knowledge, this study represents the first attempt to develop ML prediction models for CE using low-invasive patient data. We developed five comparable prediction models, each incorporating different parameters derived from gynecological history, ultrasound examination, and laboratory data. Although no significant differences were found between the models, Models 2 and 5 were identified as the most promising for clinical application due to their superior performance. These models demonstrate better recall and precision metrics and require assessment of only 6–7 risk features (with most being routine measurements) for implementation.

As we have previously described, the establishment of CE is challenging due to the lack of standardization in the diagnostic approach and the heterogeneity of clinical signs. Immunohistochemical examination for CD138+ cells requires clinical evidence, and the invasiveness and painfulness of the biopsy procedure are equally important factors since they affect quality of life and consent for examination of patients [6]. Building upon our previous research findings and incorporating published data regarding clinical associations with CE, we hypothesized that a comprehensive evaluation of potential risk factors for CE could lead to the development of a diagnostic algorithm that would not require invasive procedures. Multi-parameter modeling based on ML offers a significant advancement over traditional biomarker screening methodologies. While conventional approaches rely on univariate analyses employing statistical tests, they often fall short in capturing the complexity of biological systems [22]. According to our results, we identified two predictive models (Model 2 and Model 5) as the most promising for clinical application. Model 2 included seven features: ADIPOQ, 17-OH, T, IL-8, E2, CRP, and VAT. Model 5 included five features: SHBG, FSH, CRP, leptin/ADIPOQ ratio, and endometrial thickness. All of these parameters can be divided into four relative groups: hormones (17-OH, T, SHBG, E2, and FSH), adipokines (ADIPOQ, leptin/ADIPOQ ratio), proinflammatory markers (IL-8, CRP), and clinical features (VAT, endometrial thickness). Interestingly, both models include features that reflected reproductive function and inflammatory and metabolic state. This can mean that CE is a condition caused not only by infection, but also by hormonal and metabolic dysregulation.

The role of 17-OH in inflammation remains unclear. This hormone is primarily elevated in response to stress or due to 21-hydroxylase deficiency. Notably, elevated levels of 17-OH have been observed in patients with ankylosing spondylitis [23], suggesting a potential involvement of this hormone in autoimmune inflammatory responses. Conversely, treatment with 17-OH caproate has been shown to reduce the rate of recurrent preterm delivery in pregnant women [24]. This effect may be attributed to its regulatory influence on immune cells within the endometrium. According to our results, elevated 17-OH is associated with the risk of CE. This finding necessitates further research.

T is an important hormone for the endometrium. Androgen receptors are predominantly expressed in endometrial stromal cells, while T participates in the regulation of endometrial decidualization and prevention of oxidative stress. It is known that HA is associated with impaired endometrial receptivity, which leads to pregnancy loss [25]. In our study, patients with HA were excluded from the analysis. Moreover, low T levels were identified as a risk factor for CE. Additionally, elevated SHBG levels were also found to be a risk factor for CE, indicating the involvement of androgens in the development of endometrial inflammation. It is noteworthy that some studies have demonstrated an association of SHBG with metabolic disorders and low-grade inflammation in patients with PCOS [26]. Given the contradictory nature of the existing results, further research is required to assess the contribution of both elevated and reduced androgen levels to the development of CE.

E2 is a key regulator of endometrial function. Low E2 is associated with RIF in the first trimester [27]. However, there is limited evidence from studies showing that women with recurrent pregnancy loss may exhibit elevated basal levels of E2 [28]. Khan et al. (2015) showed in vitro promotion of pelvic inflammation (elevated levels of IL-6 and TNFα) by E2 and LPS in eutopic/ectopic endometrial stromal cells of women with endometriosis [29]. Elevated E2 can be a risk factor for endometrial inflammation. This is reflected in our results, which indicate that elevated E2 levels contribute to the CE.

FSH is a gonadotropin that is synthesized and secreted from the anterior pituitary gland. This hormone stimulates the growth and maturation of oocytes and promotes estradiol production in granulosa cells [30]. According to some authors, FSH could act on the endometrial aromatase, inducing the production of local estrogen that may confer endometrial receptivity during the period of implantation or influence estrogen-dependent epithelial production of factors related to embryo implantation [31]. There is no convincing evidence of the effect of FSH level on endometrial inflammation. Hagag et al. (2024) found significantly higher FSH in women with CE than in women without CE. But most of the women with CE were perimenopausal [32]. In our study, a low serum FSH level was associated with a higher risk of CE in premenopausal women. Further research is required to determine the contribution of FSH to the development of endometrial inflammation.

Adiponectin is an adipokine that participates in metabolism regulation and is essential for the normal functioning of the reproductive system [33]. The expression of adiponectin receptors is reduced in the endometrium of women with infertility [34]. Additionally, adiponectin plays an important anti-inflammatory role and regulates energy metabolism in the endometrium. On the other hand, a proinflammatory effect of adiponectin has been identified in autoimmune diseases [35]. Moreover, elevated levels of adiponectin have been associated with an increased risk of cardiovascular diseases [36,37]. This paradox can be explained by the impairment of liver function in such patients, since adiponectin is metabolized in the liver [38]. Thus, elevated adiponectin levels may reflect the presence of metabolic disorders associated with liver damage.

Leptin, being an adipokine and one of the key regulators of energy metabolism, is capable of stimulating the proliferation and apoptosis of endometrial epithelial cells, influencing endometrial receptivity, the uterine immune system, and endometrial decidualization [39]. In our previous study, we found a negative association between serum leptin levels and the presence of CE in women of reproductive age [17]. Upon further detailed analysis, we established that women with CE who were overweight or obese exhibited significantly lower serum leptin concentrations compared to patients without CE but who were overweight/obese [11]. These findings contradict the existing data that leptin is primarily a proinflammatory factor [40]. However, the prediction models obtained through our research also indicate that low leptin/ADIPOQ ratios are associated with CE in women with normal weight or who are overweight. In Model 2, an increase in VAT was also associated with CE, which contradicts the results obtained for adiponectin and leptin. To resolve these contradictions, further research is required, including a search for other factors that could have influenced the results.

CRP is the main marker of systemic low-grade inflammation in patients with metabolic and cardiovascular disorders [41]. On the other hand, Arefi et al. (2010) have reported higher pregnancy rates in women with elevated CRP levels on the transfer day of in vitro fertilization compared with women who had lower CRP levels [42]. Our predictive model showed a reverse association between CRP level and risk of CE. Interestingly, CRP concentration is inversely correlated with E2 during the menstrual cycle [43]. And we also found a higher risk of CE in the background of elevated E2. In this context, the observed pattern can be explained by a compensatory increase in estradiol levels in response to endometrial inflammation. However, validation of this hypothesis requires further research.

IL-8 is a proinflammatory cytokine that participates in the regulation of decidualization and vascularization of the endometrium. The studies investigating the level of IL-8 in endometrium or serum in the background of reproductive disorders are contradictory [44]. In our previous study, we found elevated serum IL-1 and IL-1/TNFα ratio, but not IL-8, in women with CE [10]. In our current research, low IL-8 is associated with a higher risk of CE.

Thin endometrium is one of the US’s findings of endometritis. Endometrial stromal thickening was found in 50–67% women with CE and recurrent pregnancy loss or RIF [45]. This finding is consistent with our data that low endometrial thickness is a risk factor for CE.

Our study has several limitations. First, the small sample size may have impacted the overall quality of the models, particularly affecting the contribution of rare features such as miscarriage rates and missed abortions. Second, it should be noted that this study has a retrospective design, originating from an epidemiology study focused on PCOS prevalence rather than CE. Third, we did not take into account the presence or absence of autoimmune conditions in women due to the lack of this information in the database. Consequently, CE diagnosis was not established for all participants. The developed prediction models are applicable exclusively to patients without comorbidities (such as HA, obesity, hypothyroidism, etc.).

## 5. Conclusions

This is the first study aimed at developing a new approach to diagnosing CE using machine learning tools. We utilized medical history, clinical, ultrasound, and laboratory parameters to construct predictive models of CE. Five models with various combinations of direct and indirect features were obtained. Although all five models demonstrated comparable performance across all metrics, we concluded that Models 2 and 5 show the most promise for further development of a low-invasive approach to diagnosing CE in premenopausal women. These models demonstrate better, though not statistically significant, recall and precision metrics. Moreover, implementing these models requires assessment of only 6–7 risk features, most of which are routine measurements.

It is worth noting that both the most promising models (Model 2 and Model 5) included features associated not only with inflammation but also with metabolic and endocrine disorders. This finding may suggest that the pathogenesis of CE involves not only infection and endometrial injury factors but also the patient’s hormonal-metabolic status. However, further research is required to validate this hypothesis and exclude the influence of confounding factors that were not accounted for in our study.

## Figures and Tables

**Figure 1 diagnostics-15-02929-f001:**
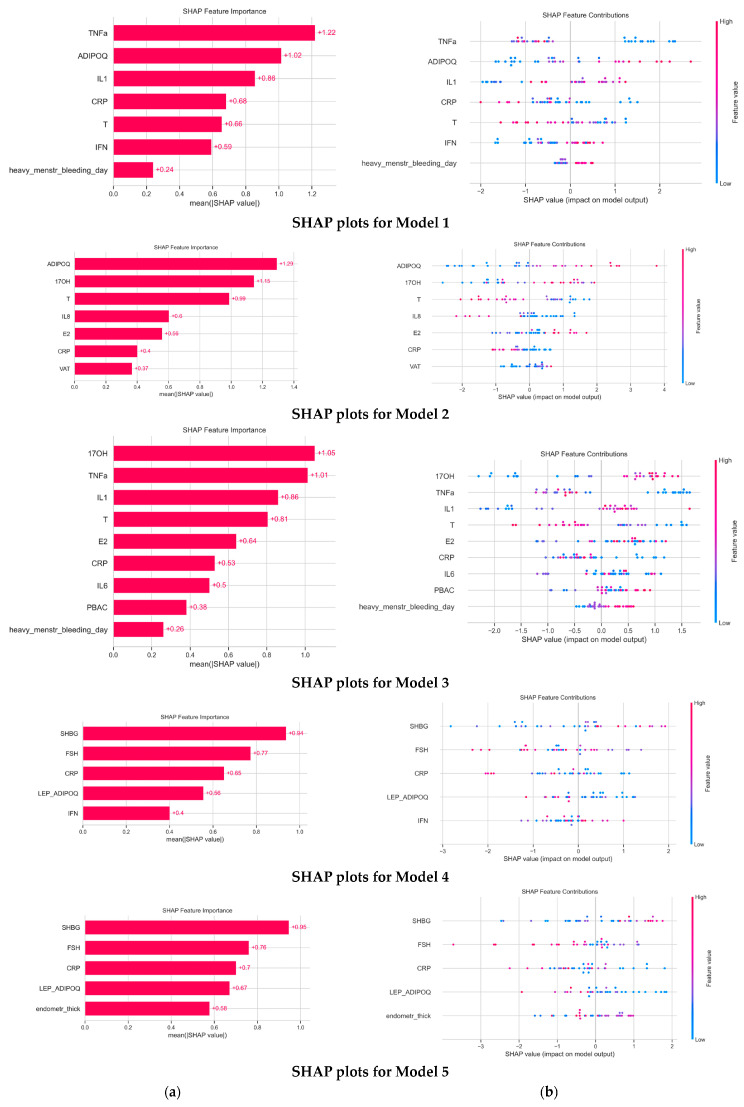
**The impact of low-invasive indicators on each CE prediction model.** (**a**) Average impact on model output magnitude and (**b**) SHAP value.

**Figure 2 diagnostics-15-02929-f002:**
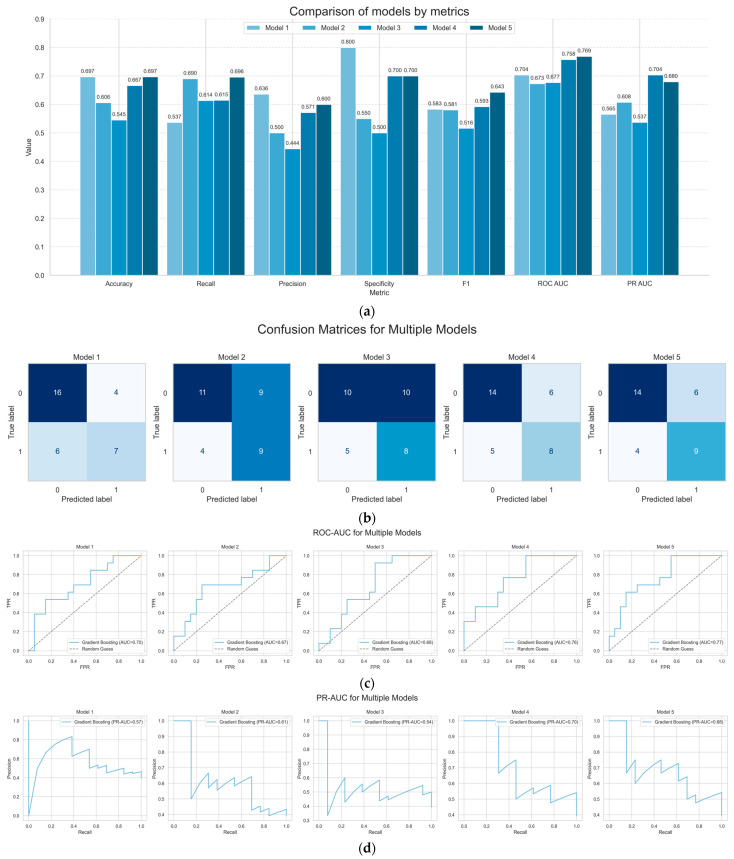
**Model comparison by metrics.** (**a**) Model comparison by accuracy, recall, precision, specificity, F1, ROC AUC, and PR AUC; (**b**) confusion matrices for models; Models 2 and 4 have the best proportion of correctly identified positive cases, but the differences are not significant (0—control; 1—CE); (**c**) PR-AUC of models; there are no significant differences between the models; (**d**) precision–recall curves of the models; there are no significant differences between the models.

**Table 1 diagnostics-15-02929-t001:** Clinical and laboratory characteristics of premenopausal women with or without CE.

Parameter	Without CE	CE	*p*
Age, years			
M ± SD	32.92 ± 5.88	35.02 ± 6.06	0.05
Me (Q1; Q3)	33.0 (28.0; 38.0)	36.0 (32.75; 39.25)	
BMI, kg/m^2^			
M ± SD	24.5 ± 3.12	24.31 ± 3.49	0.786
Me (Q1; Q3)	24.84 (21.82; 26.77)	d24.3 (21.18; 27.24)	
VAT, %			
M ± SD	d5.31 ± 1.6	5.34 ± 1.72	0.891
Me (Q1; Q3).	6.0 (4.0; 6.25)	5.0 (4.0; 7.0)	
Number of heavy menstrual bleeding days			0.646
M ± SD	2.11 ± 1.14	2.2 ± 1.0	
Me (Q1; Q3).	2.0 (1.0; 3.0)	2.0 (1.0; 3.0)	
Endometrial polyp, n(%)	0/64 (0%)	0/44 (0%)	1.000
Uterine fibroids, n (%)	4/64 (6.25%)	4/44 (9.09%)	0.713
Spontaneous_abortions, n (%)	12/62 (19.35%)	7/43 (16.28%)	0.687
missed_abortion, n (%)	4/64 (6.25%)	0/44 (0%)	0.242
extrauterine_pregnancy, n (%)	2/62 (3.23%)	2/43 (4.65%)	1.000
C-section, n (%)	13/41 (31.71%)	5/34 (14.71%)	0.086
Endometrial thickness, mm			0.512
M ± SD	8.73 ± 2.89;	8.55 ± 2.99;	
Me (Q1; Q3).	8.5 (7.0; 11.0)	8.0 (7.0; 10.0)	
FSH, mIU/L			0.639
M ± SD	5.15 ± 2.74;	4.94 ± 2.71;	
Me (Q1; Q3).	4.85 (3.3; 6.55)	4.35 (3.38; 5.8)	
**T, ng/dL**			**0.007**
**M ± SD**	**298.68 ± 144.88;**	**225.9 ± 118.9;**	
**Me (Q1; Q3).**	**274.61 (212.0; 370.53)**	**239.64 (123.24; 299.09)**	
SHBG, nmol/L			0.082
M ± SD	84.89 ± 63.58;	97.98 ± 58.31;	
Me (Q1; Q3).	70.05 (44.05; 91.05)	83.5 (52.22; 128.25)	
Leptin, ng/mL			0.109
M ± SD	21.75 ± 17.29;	14.81 ± 8.57;	
Me (Q1; Q3).	15.95 (10.0; 26.48)	14.35 (9.07; 19.73)	
Adiponectin, ng/mL			0.055
M ± SD	12.84 ± 10.51;	18.93 ± 14.51;	
Me (Q1; Q3).	10.0 (6.9; 15.2)	14.35 (8.28; 29.1)	
**Leptin/adiponectin**			**0.009**
**M ± SD**	**2.62 ± 2.47;**	**2.06 ± 3.65;**	
**Me (Q1; Q3).**	**1.72 (0.86; 4.04)**	**0.63 (0.31; 1.88)**	
CRP, IU/L			0.084
M ± SD	2.18 ± 2.52;	1.83 ± 2.77;	
Me (Q1; Q3).	1.3 (0.8; 2.45)	0.8 (0.52; 2.45)	
**IL1, ng/mL**			**0.015**
**M ± SD**	**1.37 ± 1.43;**	**2.75 ± 5.23;**	
**Me (Q1; Q3).**	**0.9 (0.5; 1.9)**	**1.55 (0.95; 2.3)**	
**TNFα, ng/mL**			**0.004**
**M ± SD**	**2.65 ± 2.07;**	**2.04 ± 2.29;**	
**Me (Q1; Q3).**	**2.0 (1.5; 2.7)**	**1.3 (0.88; 1.82)**	
**IL1/TNFα**			**<0.001**
**M ± SD**	**1.67 ± 6.81;**	**1.61 ± 1.21;**	
**Me (Q1; Q3).**	**0.63 (0.28; 0.9)**	**1.16 (0.86; 2.28)**	
IFNγ, ng/mL			0.233
M ± SD	0.82 ± 0.81	1.65 ± 3.74	
Me (Q1; Q3).	0.7 (0.2; 1.0)	0.9 (0.3; 1.3)	

**Table 2 diagnostics-15-02929-t002:** The characteristics of CE prediction models.

Model	Accuracy	Recall (CI 95%)	Precision	Specificity	F1	ROC AUC	PR AUC
Model 1	0.697	0.537	0.629	0.800	0.583	0.704	0.565
		(0.25, 0.8)	(0.333, 0.909)				
Model 2	0.606	0.690	0.502	0.550	0.581	0.673	0.608
		(0.417, 0.923)	(0.267, 0.733)				
Model 3	0.545	0.614	0.439	0.500	0.516	0.677	0.537
		(0.333, 0.9)	(0.222, 0.667)				
Model 4	0.667	0.615	0.568	0.700	0.593	0.758	0.704
		(0.333, 0.9)	(0.308, 0.818)				
Model 5	0.697	0.696	0.600	0.700	0.643	0.769	0.680
		(0.438, 0.929)	(0.357, 0.833)				

PR-ROC—precision-recall AUC.

**Table 3 diagnostics-15-02929-t003:** Comparison of CE prediction models by ROC-AUC.

Model A vs. Model B	AUC (Model A)	CI 95% (Model A)	AUC (Model B)	CI 95% (Model B)	Z	*p*
Model 1 vs. Model 2	0.70	0.5170, 0.8907	0.67	0.4716, 0.8745	−0.347	0.7285
Model 1 vs. Model 3	0.70	0.5170, 0.8907	0.68	0.4916, 0.8622	−0.314	0.7537
Model 1 vs. Model 4	0.70	0.5170, 0.8907	0.76	0.5913, 0.9241	−0.605	0.5452
Model 1 vs. Model 5	0.70	0.5170, 0.8907	0.77	0.6049, 0.9336	−0.678	0.4980
Model 2 vs. Model 3	0.67	0.4716, 0.8745	0.68	0.4916, 0.8622	−0.033	0.9739
Model 2 vs. Model 4	0.67	0.4716, 0.8745	0.76	0.5913, 0.9241	−0.795	0.4269
Model 2 vs. Model 5	0.67	0.4716, 0.8745	0.77	0.6049, 0.9336	−0.771	0.4405
Model 3 vs. Model 4	0.68	0.4916, 0.8622	0.76	0.5913, 0.9241	−0.734	0.4627
Model 3 vs. Model 5	0.68	0.4916, 0.8622	0.77	0.6049, 0.9336	−0.794	0.4273
Model 4 vs. Model 5	0.76	0.5913, 0.9241	0.77	0.6049, 0.9336	−0.200	0.8415

Z—z-statistic in the DeLong test.

**Table 4 diagnostics-15-02929-t004:** Comparison of CE prediction models by PR-AUC.

Models	PR-AUC (Model A)	PR-AUC (Model B)	CI 95% Differences	*p*-Value
Model 1 vs. Model 2	0.565	0.608	−0.283, 0.265	0.828
Model 1 vs. Model 3	0.565	0.537	−0.193, 0.324	0.720
Model 1 vs. Model 4	0.565	0.704	−0.312, 0.110	0.388
Model 1 vs. Model 5	0.565	0.680	−0.279, 0.121	0.360
Model 2 vs. Model 3	0.608	0.537	−0.227, 0.370	0.604
Model 2 vs. Model 4	0.608	0.704	−0.306, 0.117	0.392
Model 2 vs. Model 5	0.608	0.680	−0.358, 0.231	0.604
Model 3 vs. Model 4	0.537	0.704	−0.444, 0.112	0.240
Model 3 vs. Model 5	0.537	0.680	−0.432, 0.157	0.372
Model 4 vs. Model 5	0.704	0.680	−0.144, 0.201	0.828

## Data Availability

The dataset used in this research is hosted in the IEEE DataPort repository: https://dx.doi.org/10.21227/7vd8-8f90 (accessed on 19 October 2025). The source code developed for data preprocessing, selection of features, model training and model evaluation, and interpretation is publicly available in the associated GitHub repository: https://doi.org/10.5281/zenodo.17347044 (accessed on 19 October 2025).

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
