# Peer review of "Personalized Low-Invasive Approach to Chronic Endometritis Evaluation in Premenopausal Women: Machine Learning-Based Modeling"

_diagnostics, 2025, doi:10.3390/diagnostics15222929_

Round 1

Reviewer 1 Report

Comments and Suggestions for Authors

Dear Authors

Thanks for innovative and valuable study.

1-Interleukin,INF and TNF are general inflammatory mediators that may be increased in many systemic inflammatory conditions, how did you find that related to chronic endometritis?

2- please explain why did design your study on a PCO papulation?

3-Did you exclude patients with underlying disease such as auto immune, inflammatory  disease and...,? If yes please add this item to the exclusion criteria..

4- In method part please describe what CSV stand for.

5-How did you explain the AUC values (0.67–0.77) which is low.

6-what will your study help to solve the problem of RIF, recurrent spontaneous abortion, and unexplained infertility in future? 

Author Response

Dear Reviewer,

We would like to express our sincere gratitude for the thorough review of our manuscript and the insightful questions provided. Your detailed analysis has significantly contributed to the improvement of our publication and enhanced its scientific value. Please find our responses below.

Comments 1: Interleukin, INF and TNF are general inflammatory mediators that may be increased in many systemic inflammatory conditions, how did you find that related to chronic endometritis?

Response 1: Thank you for the reasonable question. These cytokines are indeed recognized as markers of various types of inflammation. Moreover, our research, along with findings from several other studies, has demonstrated an association between their levels in blood serum or directly in the endometrium and the presence of chronic endometritis (CE). This is discussed in both the introduction and the discussion sections of our study:

«These findings suggest that ultrasound examination of pelvic organs and gynecological history data can also be used as part of a comprehensive approach to CE diagnosis. In our previous studies, we had identified a significant association between elevated serum interleukin (IL) 1 levels, the IL-1/tumor necrosis factor α (TNFα) ratio, and adiponectin with CE in women with normal body mass [11,12].»

«In our previously study we had found elevated serum IL-1 and IL-1/TNFα ratio, but not IL-8 in women with CE (Ievleva 2024).»

To construct the models, we utilized data from women in whom the likelihood of inflammation caused by factors other than CE was maximally excluded. Specifically, we excluded from the sample women with: obesity, biochemical HA, hypothyroidism, sexually transmitted infections and acute conditions.

By doing so, we minimized effects of confounding variables on inflammatory cytokine levels, thereby enabling us to establish the relationship between cytokine levels and the presence of CE specifically.

Comments 2: Please explain why did design your study on a PCO papulation?

Response 2: Thank you for the clarifying question. To construct predictive models CE, we utilized a sample from a previously conducted epidemiological study on the prevalence of PCOS. This sample was chosen due to its non-selective nature, which allowed it to accurately represent the population of women residing in Eastern Siberia (participants were recruited during annual medical examinations).

Importantly, this sample included not only women with PCOS but a broader population. All participants underwent a comprehensive examination, which comprised: questionnaire survey, general physical examination, gynecological examination, ultrasound examination, laboratory tests (biochemical, hormonal, immunological),additional assessments.

The comprehensive information collected on study participants enabled us to select women for analysis while accounting for all possible confounding variables to the maximum extent.

Comments 3: Did you exclude patients with underlying disease such as auto immune, inflammatory  disease and...,? If yes please add this item to the exclusion criteria..

Response 3: Thank you for reasonable question. We excluded from the study all women with sexually transmitted infections and any types of acute conditions (this information has been added to the exclusion criteria).

However, we did not take into account the presence or absence of autoimmune conditions in women due to the lack of this information in the database. This limitation has been mentioned in the study limitations section.

Comments 4: In method part please describe what CSV stand for.

Response 4: Thank you for the clarifying question. CSV stands for "Comma-Separated Values." It is a simple file format used to store tabular data, such as a spreadsheet or database. Each line in the file represents a data row, and the values within each row are separated by commas.

The link to the dataset in the IEEE DataPort is provided in the "Materials and Methods" section to ensure the full reproducibility of our study. By making the exact dataset used for training and testing our machine learning model publicly available, other researchers can replicate our experiments and verify our results.

Comments 5: How did you explain the AUC values (0.67–0.77) which is low.

Response 5: Thank you for the reasonable question. We acknowledge that an AUC of 1.0 represents a perfect classifier, but in real-world clinical applications, especially those aiming for cost-effectiveness and minimal invasiveness, the goal is often to find a clinically useful and practically implementable model, rather than a statistically perfect one.

Our methodology was explicitly designed to prioritize the selection of the most informative features while heavily weighing their economic and clinical practicality. The series of models we built using XGBoost allowed us to explore this trade-off systematically.

The two models with AUCs of 0.67 and 0.77 were selected not because they were the highest-performing in our pool, but because they offered the best balance between:

Acceptable Discriminatory Power: It is important to note that an AUC above 0.7 is generally considered acceptable, and an AUC of 0.6-0.7 may still provide use ful discriminative capacity in many medical contexts (ÇorbacıoÄŸlu, Åžeref Kerem1,,*; Aksel, Gökhan2. Receiver operating characteristic curve analysis in diagnostic accuracy studies: A guide to interpreting the area under the curve value. Turkish Journal of Emergency Medicine 23(4):p 195-198, Oct–Dec 2023. | DOI: 10.4103/tjem.tjem_182_23).

Economic Efficiency and Minimal Invasiveness: The feature sets in these models consist of variables that are low-cost and easy to collect, which is the cornerstone of our proposed diagnostic procedure.

A model with a marginally higher AUC but requiring expensive or invasive tests would defeat the core purpose of our study. Therefore, we argue that the reported AUC values, while modest, are clinically meaningful and represent a significant step towards a feasible, low-cost diagnostic tool A. M. Carrington et al., "Deep ROC Analysis and AUC as Balanced Average Accuracy, for Improved Classifier Selection, Audit and Explanation," in IEEE Transactions on Pattern Analysis and Machine Intelligence, vol. 45, no. 1, pp. 329-341, 1 Jan. 2023, doi: 10.1109/TPAMI.2022.3145392). The high practical utility and cost-effectiveness of the selected models justify their choice for further development and validation.

6-what will your study help to solve the problem of RIF, recurrent spontaneous abortion, and unexplained infertility in future? 

In the introduction section, we stated that CE is one of the most common endometrial disorders in patients with RIF, recurrent spontaneous abortion, and unexplained infertility. The developed personalized minimally invasive method for diagnosing CE, due to its greater accessibility, ease of use, and relative affordability, can increase the detection rate of CE in these patients and women at risk for CE, which will contribute to the prevention of CE complications in the form of reproductive losses through adjustment of patient treatment towards anti-inflammatory therapy and correction of endometrial function. Of course, for the practical application of this approach, it is necessary to repeat the study on a larger sample, establishing a direct relationship between the parameters included in the model and inflammation indicators directly in the endometrium, which we mentioned in the limitations and also added to the conclusion of the study.

Reviewer 2 Report

Comments and Suggestions for Authors

I reviewed the article titled (Personalized Low-Invasive Approach to Chronic Endometritis Evaluation in Premenopausal Women: Machine Learning-Based Modeling)

1 - in introduction: 

What specific machine learning techniques were used to develop the CE prediction model? How will the new diagnostic approach change the current clinical practices for CE evaluation? What are the potential implications of accurately diagnosing chronic endometritis for the treatment of women with infertility?

2- in methodolgy : 

What specific clinical outcomes were evaluated in relation to the inclusion of CD138+ expression in endometrial stroma? How were the potential effects of confounding variables controlled for in the analysis?

3- in discussion: 

What specific mechanisms link the hormonal and metabolic factors identified in the study to the development of CE? How might the findings about 17-OH and its association with autoimmune inflammatory responses influence future research on CE? What additional risk factors for CE could be explored in future studies to enhance the diagnostic algorithm?

Author Response

We would like to express our sincere gratitude for the thorough review of our manuscript and the insightful questions provided. Your detailed analysis has significantly contributed to the improvement of our publication and enhanced its scientific value. Please find our responses below.

Comments 1:  in introduction: 

What specific machine learning techniques were used to develop the CE prediction model? How will the new diagnostic approach change the current clinical practices for CE evaluation? What are the potential implications of accurately diagnosing chronic endometritis for the treatment of women with infertility?

Response 1: Thank you for the clarifying question. Previous studies have addressed CE prediction using deep learning models, primarily convolutional neural networks (CNNs), for analyzing hysteroscopic images to link findings with histopathologic CE diagnosis (Mihara, M.; Yasuo, T.; Kitaya, K. Precision Medicine for Chronic Endometritis: Computer-Aided Diagnosis Using Deep Learning Model. Diagnostics 2023, 13, 936. https://doi.org/10.3390/diagnostics13050936; Kitaya, K.; Yasuo, T.; Yamaguchi, T. Bridging the Diagnostic Gap between Histopathologic and Hysteroscopic Chronic Endometritis with Deep Learning Models. Medicina 2024, 60, 972. https://doi.org/10.3390/medicina60060972).

The novel diagnostic approach to CE has the potential to enhance early detection of this condition before complications and reproductive losses develop. Furthermore, the identification of CE in women with infertility will enable therapeutic adjustments aimed at restoring endometrial function. We have added this information to the introduction section

Comments 2: in methodolgy : 

What specific clinical outcomes were evaluated in relation to the inclusion of CD138+ expression in endometrial stroma? How were the potential effects of confounding variables controlled for in the analysis?

Response 2: Thank you for the clarifying question. To construct the models, we utilized the database from a previously conducted population-based study. According to the study protocol, endometrial biopsy was performed on all patients who consented to the procedure, without selecting participants based on clinical symptoms of CE. We have added this information into the introduction section.

During the sample formation process, we excluded data from women whose conditions could be associated with inflammation development (obesity, HA). Additionally, women with other hormonal disorders, sexually transmitted infections, and acute conditions were excluded from the study.

Given that we utilized the maximum possible exclusion criteria for this database — signs potentially associated with inflammation — we did not perform further confounding variable control. This information has been added to the Materials and Methods section.

However, we did not use autoimmune diseases as an exclusion criterion due to the absence of this information in the database. This information has been added to the study limitations section.

Comments 3: in discussion: 

What specific mechanisms link the hormonal and metabolic factors identified in the study to the development of CE? How might the findings about 17-OH and its association with autoimmune inflammatory responses influence future research on CE? What additional risk factors for CE could be explored in future studies to enhance the diagnostic algorithm?

Response 3: Thank you for the reasonable question. Indeed, to date, there are no studies confirming a direct relationship between hormonal and metabolic disorders and the presence of CE. In the discussion section, we described potential mechanisms of association for each parameter included in the most promising models. This includes a review of currently available data regarding the relationship between these indicators and the development of inflammation, including autoimmune inflammation.

Undoubtedly, additional research is necessary, which we emphasize in the discussion. At present, we refrain from proposing any hypothesis due to the limited amount of accumulated data.

The diagnostic algorithm could be strengthened by incorporating gynecological history data, which we were unable to achieve in this study due to the small sample size and limited number of cases. This limitation has been mentioned in both the study limitations and the conclusion sections.